# REINFORCING DIFFUSION MODELS BY DIRECT GROUP PREFERENCE OPTIMIZATION

**Yihong Luo**[1]     **Tianyang Hu**[2]     **Jing Tang**[3,1*]
[1] HKUST     [2] CUHK (SZ)     [3] HKUST (GZ)

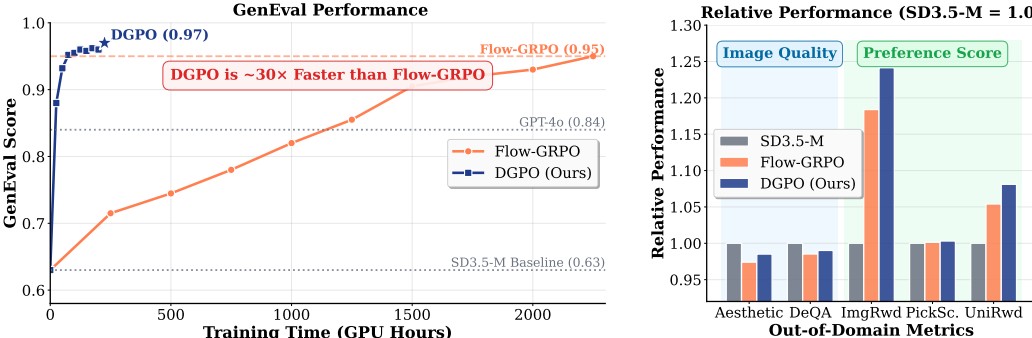

Figure 1: Our proposed **DGPO** shows a near 30 times faster training compared to Flow-GRPO on improving GenEval score (Left Figure). The notable improvement is achieved while maintaining strong performance on other out-of-domain metrics (Right Figure).

## ABSTRACT

While reinforcement learning methods such as Group Relative Preference Optimization (GRPO) have significantly enhanced Large Language Models, adapting them to diffusion models remains challenging. In particular, GRPO demands a stochastic policy, yet the most cost-effective diffusion samplers are based on deterministic ODEs. Recent work addresses this issue by using inefficient SDE-based samplers to induce stochasticity, but this reliance on model-agnostic Gaussian noise leads to slow convergence. To resolve this conflict, we propose Direct Group Preference Optimization (DGPO), a new online RL algorithm that dispenses with the policy-gradient framework entirely. DGPO learns directly from group-level preferences, which utilize relative information of samples within groups. This design eliminates the need for inefficient stochastic policies, unlocking the use of efficient deterministic ODE samplers and faster training. Extensive results show that DGPO trains around 20 times faster than existing state-of-the-art methods and achieves superior performance on both in-domain and out-of-domain reward metrics. Code is available at `https://github.com/Luo-Yihong/DGPO`.

## 1   INTRODUCTION

Reinforcement Learning (RL) has become a cornerstone for the post-training of Large Language Models (LLMs), significantly enhancing their capabilities (Ziegler et al., 2019; Ouyang et al., 2022; Bai et al., 2022). In particular, methods like Group Relative Policy Optimization (GRPO) (Shao et al., 2024) have demonstrated remarkable success in substantially improving the complex reasoning abilities of LLMs (DeepSeek-AI, 2025). However, progress in applying RL for post-training diffusion models has lagged considerably behind that of language models, leaving a significant gap in methods for aligning generative models with human preferences and complex quality metrics.

A central obstacle is the mismatch between GRPO's policy gradient-based framework (short for policy framework in the following) and the mechanics of diffusion generation. GRPO requires access to a stochastic policy to enable effective training and exploration. This requirement is naturally

---

*Corresponding Author: Jing Tang

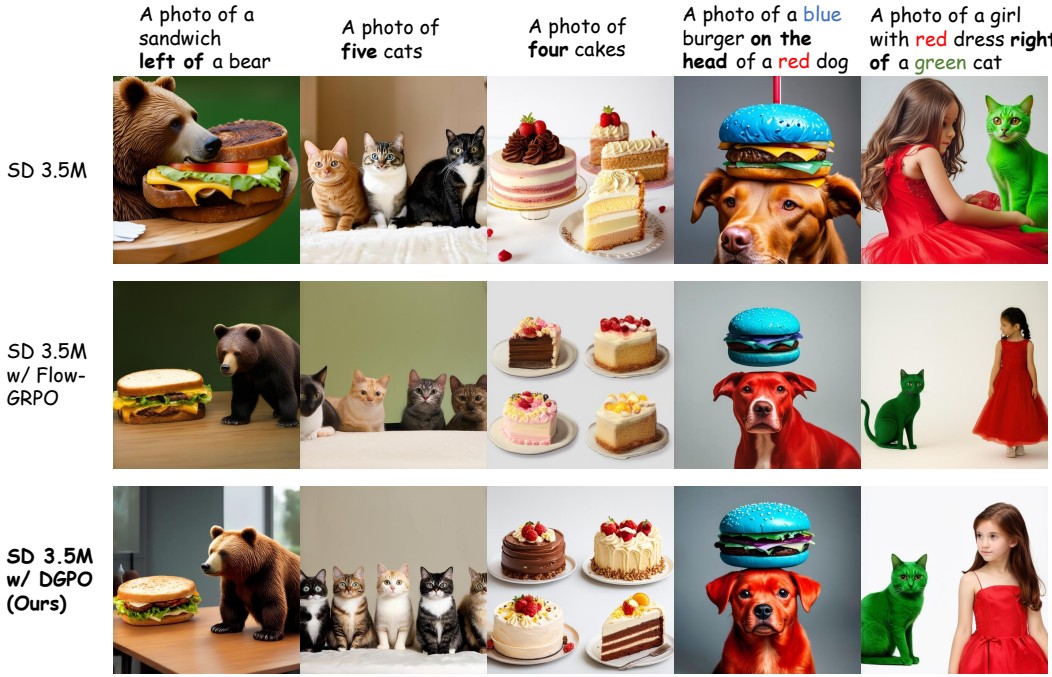

Figure 2: Qualitative comparisons of DGPO against competing methods. It can be seen that our proposed DGPO not only accurately follows the instructions, but also keeps a strong visual quality. All images are generated by the same initial noise.

met by LLMs, which inherently output a probability distribution over a vocabulary. In contrast, diffusion models predominantly rely on deterministic ODE-based samplers to strike a better balance between sample quality and cost (Song et al., 2020; Luo et al., 2025c), and thus do not naturally provide a stochastic policy. To bridge this gap, prior work has resorted to a forced adaptation: using stochastic SDE-based sampling to induce a conditional Gaussian policy suitable for GRPO's policy framework (Liu et al., 2025; Xue et al., 2025). This workaround, however, introduces severe negative consequences: (1) SDE-based rollouts are less efficient than their ODE counterparts and produce lower-quality samples under a fixed computational budget (Lu et al., 2022a;b; Bao et al., 2022; Song et al., 2020); (2) The policy's stochasticity comes from model-agnostic Gaussian noise, which provides a weak learning signal and results in slow convergence; and (3) Training is performed over the entire sampling trajectory, making each iteration computationally expensive and time-consuming.

We argue that the practical success of GRPO stems less from its policy-gradient formulation, and more from its ability to utilize fine-grained relative preference information within group. Based on the insight, an ideal RL method for diffusion models should be capable of leveraging this powerful group-level information while dispensing with the need for a stochastic policy and its associated negative effects. To this end, we introduce **D**irect **G**roup **P**reference **O**ptimization (**DGPO**), a new online RL method tailored to diffusion models. DGPO circumvents the policy-gradient framework entirely, instead optimizing the model by directly learning from the group-level preference between a set of "good" samples and a set of "bad" samples. Concretely, for each prompt, we generate $G$ samples using efficient ODE-based rollouts, partition them into positive and negative groups, and directly optimize the model by maximizing the likelihood of these group-wise preferences. Conceptually, DGPO can be understood as a natural extension of Direct Preference Optimization (DPO) (Wallace et al., 2024) that incorporates group-wise information, and as a diffusion-native re-imagination of GRPO.

This proposed methodology allows us to bypass the dependency on a stochastic policy, which yields several benefits: (1) **Efficient Sampling and Learning**: by using high-fidelity ODE samplers, DGPO learns from higher-quality rollouts, leading to more effective learning. (2) **Efficient Convergence**: optimization is directly guided by group-level preferences rather than inefficient model-agnostic random exploration, leading to faster convergence. (3) **Efficient Training**: our approach

avoids training on the entire sampling trajectory, notably reducing the computational cost of each training iteration. Together, these advantages establish DGPO as a highly efficient and powerful online RL algorithm for diffusion models. Our extensive experiments show that DGPO achieves around 20× faster training than prior state-of-the-art Flow-GRPO (Liu et al., 2025), while delivering superior performance on both in-domain and out-of-domain metrics. Most notably, on the challenging GenEval benchmark (Ghosh et al., 2023), DGPO trains nearly 30× faster than Flow-GRPO and boosts the base model's performance from 63% to 97% (Fig. 1). These compelling results demonstrate DGPO's potential as a powerful technique for aligning diffusion models.

## 2 PRELIMINARIES

**Diffusion Models (DMs)**   DMs (Sohl-Dickstein et al., 2015; Ho et al., 2020) define a forward diffusion mechanism that progressively introduces Gaussian noise to input data $\mathbf{x}$ across $T$ sequential timesteps. The forward process follows the distribution $q(\mathbf{x}_t|\mathbf{x}) \triangleq \mathcal{N}(\mathbf{x}_t; \alpha_t\mathbf{x}, \sigma_t^2\mathbf{I})$, where the hyperparameters $\alpha_t$ and $\sigma_t$ control the noise scheduling strategy. At each timestep, noisy samples are obtained by $\mathbf{x}_t = \alpha_t\mathbf{x} + \sigma_t\epsilon$, where $\epsilon \sim \mathcal{N}(\mathbf{0}, \mathbf{I})$. The parameterized reversed diffusion process is defined by: $p_\theta(\mathbf{x}_{t-1}|\mathbf{x}_t) \triangleq \mathcal{N}(\mathbf{x}_{t-1}; \mu_\theta(\mathbf{x}_t, t), \eta_t^2\mathbf{I})$. The model's neural network $f_\theta$ is learned by denoising $\mathbb{E}_{\mathbf{x}, \epsilon, t}\lambda_t||f_\theta(\mathbf{x}_t, t) - \mathbf{x}||_2^2$. We note that the flow matching and DMs are equivalent in the context of diffusing by Gaussian noise (Gao et al., 2025).

**Reward Modeling**   Given ranked pairs generated from certain conditioning $\mathbf{x}_0^w \succ \mathbf{x}_0^l|\mathbf{c}$, where $\mathbf{x}_0^w$ and $\mathbf{x}_0^l$ denote the "better" and "worse" samples. The Bradley-Terry (BT) model formulates the preferences as:

$$p_{\text{BT}}(\mathbf{x}_0^w \succ \mathbf{x}_0^l|\mathbf{c}) = \sigma(r(\mathbf{c}, \mathbf{x}_0^w) - r(\mathbf{c}, \mathbf{x}_0^l)) \tag{1}$$

where $\sigma(\cdot)$ denotes the sigmoid function. A network $r_\phi$ that models reward can be trained by maximum likelihood as follows:

$$L_{\text{BT}}(\phi) = -\mathbb{E}_{\mathbf{c}, \mathbf{x}_0^w, \mathbf{x}_0^l}\left[\log\sigma\left(r_\phi(\mathbf{c}, \mathbf{x}_0^w) - r_\phi(\mathbf{c}, \mathbf{x}_0^l)\right)\right] \tag{2}$$

**RLHF**   RLHF typically aims to optimize a conditional density $p_\theta(\mathbf{x}_0|\mathbf{c})$ to maximize a underlying reward $r(\mathbf{c}, \mathbf{x}_0)$ while staying close to a reference distribution $p_{\text{ref}}$ via KL regularization, i.e.,

$$\max_{p_\theta} \mathbb{E}_{\mathbf{c}, \mathbf{x}_0 \sim p_\theta(\mathbf{x}_0|\mathbf{c})}\left[r(\mathbf{c}, \mathbf{x}_0)\right] - \beta\text{KL}\left[p_\theta(\mathbf{x}_0|\mathbf{c})\|p_{\text{ref}}(\mathbf{x}_0|\mathbf{c})\right] \tag{3}$$

where the hyperparameter $\beta$ controls strength of regularization.

**GRPO Objective (Shao et al., 2024)**   The RLHF objective in Eq. (3) can be optimized by policy-based learning (we omit the KL term and clip term hereinafter for brevity):

$$\max_{p_\theta} \mathbb{E}_{(\mathbf{x}_0, \mathbf{x}_1, \cdots, \mathbf{x}_T) \sim p_{\theta_{\text{old}}}(\cdot|\mathbf{c})} \sum_{k=0}^{T} \frac{p_\theta(\mathbf{x}_{k+1}|\mathbf{x}_k, \mathbf{c})}{p_{\theta_{\text{old}}}(\mathbf{x}_{k+1}|\mathbf{x}_k, \mathbf{c})} A(\mathbf{x}_{k+1}), \tag{4}$$

where $A(\mathbf{x}_{k+1})$ denotes the advantage of $\mathbf{x}_{k+1}$, which can be directly computed by reward $r(\mathbf{x}_{k+1})$, or introduce additional value model for reducing variance.

The GRPO proposes to sample a group of outputs for each prompt $\mathbf{c}$ from the old policy, then compute the advantage of each sample by normalization among groups, i.e., $A_i = (r_i - \text{mean}(\{r_1, r_2, \cdots, r_G\}))/\text{std}(\{r_1, r_2, \cdots, r_G\})$. The policy learning requires that the transition between $\mathbf{x}_k$ and $\mathbf{x}_{k+1}$ follows a stochastic distribution. To meet this requirement for a stochastic policy, recent works (Liu et al., 2025; Xue et al., 2025) employ a stochastic SDE for sampling, rather than the more efficient deterministic ODE. However, the SDE itself is less effective in sampling high-quality samples with insufficient steps. Besides, the policy-based method requires performing training on the whole trajectory, which further leads to slow training. More importantly, unlike LLMs, which directly output distribution, *the stochasticity in DM's policy comes from model-agnostic Gaussian Noise. This makes the stochastic exploration rely on the model-agnostic Gaussian noise, which is extremely inefficient in high-dimensional space.*

**DPO Objective (Rafailov et al., 2024)** The unique global optimal density $p_\theta^*$ of the RLHF objective (Eq. (3)) is given by:

$$p_\theta^*(\mathbf{x}_0|\mathbf{c}) = p_{\text{ref}}(\mathbf{x}_0|\mathbf{c}) \exp\left(r(\mathbf{c}, \mathbf{x}_0)/\beta\right) / Z(\mathbf{c}) \tag{5}$$

where $Z(\mathbf{c}) = \sum_{\mathbf{x}_0} p_{\text{ref}}(\mathbf{x}_0|\mathbf{c}) \exp\left(r(\mathbf{c}, \mathbf{x}_0)/\beta\right)$ is a intractable partition function. We can compute the reward function as follows:

$$r(\mathbf{c}, \mathbf{x}_0) = \beta \log \frac{p_\theta^*(\mathbf{x}_0|\mathbf{c})}{p_{\text{ref}}(\mathbf{x}_0|\mathbf{c})} + \beta \log Z(\mathbf{c}) \tag{6}$$

After obtaining the parameterization of the reward function, the DPO optimizes the models by the reward learning objective in Eq. (2):

$$L_{\text{DPO}}(\theta) = -\mathbb{E}_{\mathbf{c}, \mathbf{x}_0^w, \mathbf{x}_0^l}\left[\log \sigma\left(\beta \log \frac{p_\theta(\mathbf{x}_0^w|\mathbf{c})}{p_{\text{ref}}(\mathbf{x}_0^w|\mathbf{c})} - \beta \log \frac{p_\theta(\mathbf{x}_0^l|\mathbf{c})}{p_{\text{ref}}(\mathbf{x}_0^l|\mathbf{c})}\right)\right] \tag{7}$$

Diffusion DPO (Wallace et al., 2024) has adapted DPO to Diffusion models by defining reward over the diffusion paths $\mathbf{x}_{0:T}$, which does not require a stochastic policy. *However, it strictly relies on pairwise samples for optimization due to the intrinsic restriction of the intractable partition $Z(\mathbf{c})$, preventing the use of the fine-grained preference information of each sample.*

## 3 METHOD

We believe that the key to GRPO's success lies in its ability to utilize fine-grained relative preference information within groups. However, existing GRPO-style methods (Liu et al., 2025; Xue et al., 2025) require using an inefficient stochastic policy. Although existing DPO-style methods (Wallace et al., 2024) provide a framework without the need for a stochastic policy, they require performing training on pairwise samples to eliminate the intractable partition $Z(\mathbf{c})$. To this end, we propose Direct Group Preference Optimization (DGPO), which eliminates the inefficient stochastic policy and allows us to directly optimize inter-group preferences without concerning ourselves with an intractable partition, leveraging fine-grained reward information to significantly improve training efficiency. The pseudo code of DGPO is summarized in Algorithm 1.

**Problem Setup** Let $p_{\text{ref}}$ denote a pre-trained reference diffusion model with parameter $\theta_{\text{ref}}$. We have a dataset of conditions $\mathcal{D}_c = \{\mathbf{c}_{i=1}^N\}$ and a reward function $r_\phi(\cdot, \cdot) : \mathcal{X} \times \mathcal{C} \to \mathbb{R}$ that evaluates the quality of generated samples $\mathbf{x} \in \mathcal{X}$ given condition $\mathbf{c} \in \mathcal{C}$. Our goal is to enhance a diffusion model $p_\theta$, initialized from $p_{\text{ref}}$, according to the reward signal. At each training iteration, we use an online model $p_{\theta^-}$ to generate a group of samples conditioned on $c \in \mathcal{D}_c$, where $\theta^-$ can be set as the current parameters $\theta$ or an exponential moving average (EMA) version of previous $\theta$'s. These generated samples form a dataset $\mathcal{D} = \{(\mathcal{G}_i = \{\mathbf{x}_{k=1}^G\}, \mathbf{c}_i) \mid \mathbf{x}_k \sim p_{\theta^-}(\cdot|\mathbf{c}_i)\}$, which are then evaluated by the reward function to provide reward signals for splitting positive or negative groups.

### 3.1 DIRECT GROUP PREFERENCE OPTIMIZATION

In order to leverage relative information within groups, we propose directly learn the group-level preferences using the Bradley-Terry model via maximum likelihood:

$$\max_\theta \mathbb{E}_{(\mathcal{G}^+, \mathcal{G}^-, c) \sim \mathcal{D}} \log p(\mathcal{G}^+ \succ \mathcal{G}^- |\mathbf{c}) = \mathbb{E}_{(\mathcal{G}^+, \mathcal{G}^-, c) \sim \mathcal{D}} \log \sigma(R_\theta(\mathcal{G}^+|\mathbf{c}) - R_\theta(\mathcal{G}^-|\mathbf{c})) \tag{8}$$

where $\mathcal{G}^+$ and $\mathcal{G}^-$ represent positive and negative groups respectively, with $\mathcal{G}^+ \cup \mathcal{G}^- = \mathcal{G}$, and $\mathcal{G} = \{\mathbf{x}_0^1, \cdots, \mathbf{x}_0^G\}$ being the complete group of samples for conditioning $\mathbf{c}$. Intuitively, the objective in Eq. (8) can leverage fine-grained preference information of each sample within groups with appropriate parameterization.

Therefore, we propose parameterizing the group-level reward as a weighted sum of rewards $r_\theta(\mathbf{c}, \mathbf{x}_0)$ for each sample within the group:

$$R_\theta(\mathcal{G}|\mathbf{c}) = \sum_{\mathbf{x}_0 \in \mathcal{G}} w(\mathbf{x}_0) \cdot r_\theta(\mathbf{c}, \mathbf{x}_0), \tag{9}$$

where $\omega$ controls the "importance level" of the sample within the group. The parameterization of group-level reward can reflect the fine-grained information of each sample. And the reward of single

---

**Algorithm 1** Direct Group Preference Optimization (DGPO)

**Require:** Diffusion model $f_\theta$, Reference model $f_{\text{ref}}$, Reward model $r_\phi$, Group size $G$, Hyperparameter $\beta$, Minimum training timestep $t_{\min}$, Learning rate $\eta$, Iterations $N$, EMA decay $\mu$ (optional).
**Ensure:** Optimized model $f_\theta$.
1: **for** $n \leftarrow 1$ **to** $N$ **do**
2:    # Sample conditioning and generate group
3:    Sample conditioning $\mathbf{c} \sim \mathcal{D}_c$
4:    Generate group $\mathcal{G} = \{\mathbf{x}_0^1, ..., \mathbf{x}_0^G\}$ by sampling from $p_{\theta^-}(\cdot|\mathbf{c})$
5:    # Compute advantages
6:    $\{r_i\} \leftarrow \{r_\phi(\mathbf{c}, \mathbf{x}_0^i)\}_{i=1}^G$
7:    $A_i \leftarrow \frac{r_i - \text{mean}(\{r_j\})}{\text{std}(\{r_j\})}$ for all $i$
8:    # Partition into positive and negative groups
9:    $\mathcal{G}^+ \leftarrow \{\mathbf{x}_0^i : A_i > 0\}$ and $\mathcal{G}^- \leftarrow \{\mathbf{x}_0^i : A_i \leq 0\}$
10:   # Compute DGPO loss
11:   Sample $t \sim \mathcal{U}[t_{\min}, T]$, $\epsilon \sim \mathcal{N}(0, I)$
12:   $\mathbf{x}_t^i \leftarrow \alpha_t \mathbf{x}_0^i + \sigma_t \epsilon$ for all $i$
13:   Compute $\mathcal{L}_{\text{DGPO}}$ by Eq. (17)
14:   Update $\theta \leftarrow \theta - \eta \nabla_\theta \mathcal{L}_{\text{DGPO}}$
15:   Update $\theta^- \leftarrow \theta$ or $\theta^- \leftarrow \mu\theta^- + (1 - \mu)\theta$
16: **end for**

---

sample can be parameterized by following Eq. (6) and Diffusion-DPO (Wallace et al., 2024):

$$
\begin{aligned}
r_\theta(\mathbf{c}, \mathbf{x}_0) &= \beta \mathbb{E}_{p_\theta(\mathbf{x}_{1:T}|\mathbf{x}_0)} \log \frac{p_\theta(\mathbf{x}_{0:T}|\mathbf{c})}{p_{\text{ref}}(\mathbf{x}_{0:T}|\mathbf{c})} + \beta \log Z(\mathbf{c}), \\
&\approx \beta \mathbb{E}_{q(\mathbf{x}_{1:T}|\mathbf{x}_0)} \log \frac{p_\theta(\mathbf{x}_{0:T}|\mathbf{c})}{p_{\text{ref}}(\mathbf{x}_{0:T}|\mathbf{c})} + \beta \log Z(\mathbf{c})
\end{aligned}
\tag{10}
$$

where $Z(\mathbf{c})$ is a intractable partition function. We note that since sampling from the inversion chain $p_\theta(\mathbf{x}_{1:T}|x_0)$ is expensive, the forward diffusion $q(\mathbf{x}_{1:T}|\mathbf{x}_0)$ has been utilized as an approximation in practice (Wallace et al., 2024).

By combining Eq. (9) and Eq. (8), we can derive the desired training objective:

$$
\begin{aligned}
L(\theta) &= -\mathbb{E}_{(\mathcal{G}^+, \mathcal{G}^-, c) \sim \mathcal{D}} \log \sigma \Big( \sum_{\mathbf{x}_0 \in \mathcal{G}^+} \mathbb{E}_{q(\mathbf{x}_{1:T}|\mathbf{x}_0)} \beta w(\mathbf{x}_0) [\log \frac{p_\theta(\mathbf{x}_{0:T}|\mathbf{c})}{p_{\text{ref}}(\mathbf{x}_{0:T}|\mathbf{c})} + Z(\mathbf{c})] \\
&\quad - \sum_{\mathbf{x}_0 \in \mathcal{G}^-} \mathbb{E}_{q(\mathbf{x}_{1:T}|\mathbf{x}_0)} \beta w(\mathbf{x}_0) \cdot [\log \frac{p_\theta(\mathbf{x}_{0:T}|\mathbf{c})}{p_{\text{ref}}(\mathbf{x}_{0:T}|\mathbf{c})} + Z(\mathbf{c})]] \Big) \\
&= -\mathbb{E}_{(\mathcal{G}^+, \mathcal{G}^-, c) \sim \mathcal{D}} \log \sigma \Big( \beta [\mathbb{E}_{q(\mathbf{x}_{1:T}|\mathbf{x}_0)} \sum_{\mathbf{x}_0 \in \mathcal{G}^+} w(\mathbf{x}_0) \cdot \log \frac{p_\theta(\mathbf{x}_{0:T}|\mathbf{c})}{p_{\text{ref}}(\mathbf{x}_{0:T}|\mathbf{c})} \\
&\quad - \sum_{\mathbf{x}_0 \in \mathcal{G}^-} \mathbb{E}_{q(\mathbf{x}_{1:T}|\mathbf{x}_0)} w(\mathbf{x}_0) \log \frac{p_\theta(\mathbf{x}_{0:T}|\mathbf{c})}{p_{\text{ref}}(\mathbf{x}_{0:T}|\mathbf{c})} + \sum_{\mathbf{x}_0 \in \mathcal{G}^+} w(\mathbf{x}_0) Z(\mathbf{c}) - \sum_{\mathbf{x}_0 \in \mathcal{G}^-} w(\mathbf{x}_0) Z(\mathbf{c})] \Big)
\end{aligned}
\tag{11}
$$

A remaining crucial challenge is that the partition function $Z(\mathbf{c})$ is intractable for training. We have to carefully select an appropriate weighting $w_i$ for each sample to eliminate the intractable partition function $Z(\mathbf{c})$. Generally speaking, a good weighting strategy should satisfy the following:

- Larger weights correspond to better samples in $\mathcal{G}^+$ and worse samples in $\mathcal{G}^-$.
- The weights satisfy: $\sum_{\mathbf{x}_0 \in \mathcal{G}^+} w(\mathbf{x}_0) = \sum_{\mathbf{x}_0 \in \mathcal{G}^-} w(\mathbf{x}_0)$, such that $\sum_{\mathbf{x}_0 \in \mathcal{G}^+} w(\mathbf{x}_0) Z(\mathbf{c}) - \sum_{\mathbf{x}_0 \in \mathcal{G}^-} w(\mathbf{x}_0) Z(\mathbf{c})) = 0$ for eliminating the intractable $Z(\mathbf{c})$.

## 3.2 ADVANTAGE-BASED WEIGHT DESIGN

We propose using advantage-based weights derived from GRPO-style normalization to address the aforementioned issues. Given a group $\mathcal{G} = \mathbf{x}_0^1, \mathbf{x}_0^2, ..., \mathbf{x}_0^G$ with corresponding rewards $r_1, r_2, ..., r_G$,

we compute advantages:

$$A(\mathbf{x}_0^i) = \frac{r_i - \text{mean}(\{r_j\}_{j=1}^G)}{\text{std}(\{r_j\}_{j=1}^G)} \tag{12}$$

We then partition the group based on advantages:

$$\mathcal{G}^+ = \{\mathbf{x}_0^i : A(\mathbf{x}_0^i) > 0\}, \quad \mathcal{G}^- = \{\mathbf{x}_0^i : A(\mathbf{x}_0^i) \leq 0\}. \tag{13}$$

And we set weights as:

$$w(\mathbf{x}_0) = |A(\mathbf{x}_0)| \tag{14}$$

This choice ensures $\sum_{\mathbf{x}_0 \in \mathcal{G}^+} w(\mathbf{x}_0) = \sum_{\mathbf{x}_0 \in \mathcal{G}^-} w(\mathbf{x}_0)$ due to the zero-mean property of the normalized advantages. It also dynamically assigns larger weights to samples that deviate more from the average, which enables the model to more effectively learn relative preference relationships. More importantly, this weighting turns the objective in Eq. (11) to:

$$L(\theta) = -\mathbb{E}_{(\mathcal{G}^+, \mathcal{G}^-, c) \sim \mathcal{D}} \log \sigma(\beta[\sum_{\mathbf{x}_0 \in \mathcal{G}^+} \mathbb{E}_{q(\mathbf{x}_{1:T}|\mathbf{x}_0)} w(\mathbf{x}_0) \log \frac{p_\theta(\mathbf{x}_{0:T}|\mathbf{c})}{p_{\text{ref}}(\mathbf{x}_{0:T}|\mathbf{c})}$$
$$- \sum_{\mathbf{x}_0 \in \mathcal{G}^-} w(\mathbf{x}_0) \mathbb{E}_{q(\mathbf{x}_{1:T}|\mathbf{x}_0)} \log \frac{p_\theta(\mathbf{x}_{0:T}|\mathbf{c})}{p_{\text{ref}}(\mathbf{x}_{0:T}|\mathbf{c})}]). \tag{15}$$

By using Jensen's inequality and the convexity of $-\log\sigma$, we can move the expectation outside:

$$L(\theta) \leq -\mathbb{E}_{(\mathcal{G}^+, \mathcal{G}^-, c) \sim \mathcal{D}} \mathbb{E}_{t, q(\mathbf{x}_t|\mathbf{x}_0)} \log \sigma(\sum_{\mathbf{x}_0 \in \mathcal{G}^+} w(\mathbf{x}_0)\beta T \mathbb{E}_{q(\mathbf{x}_{t-1}|\mathbf{x}_t, \mathbf{x}_0)} \log \frac{p_\theta(\mathbf{x}_{t-1}|\mathbf{x}_t, \mathbf{c})}{p_{\text{ref}}(\mathbf{x}_{t-1}|\mathbf{x}_t, \mathbf{c})}$$
$$- \sum_{\mathbf{x}_0 \in \mathcal{G}^-} w(\mathbf{x}_0)\beta T \mathbb{E}_{q(\mathbf{x}_{t-1}|\mathbf{x}_t, \mathbf{x}_0)} \log \frac{p_\theta(\mathbf{x}_{t-1}|\mathbf{x}_t, \mathbf{c})}{p_{\text{ref}}(\mathbf{x}_{t-1}|\mathbf{x}_t, \mathbf{c})}), \tag{16}$$

where $\mathcal{G} = \mathcal{G}^+ \cup \mathcal{G}^-$. We note that to reduce the variance, the sampled noise $\epsilon$ is shared among samples within the same complete groups. By some simplification, we obtain our final training objective of the proposed DGPO:

$$L_{\text{DGPO}}(\theta) \triangleq -\mathbb{E}_{(\mathcal{G}^+, \mathcal{G}^-, c) \sim \mathcal{D}} \mathbb{E}_{t, q(\mathbf{x}_t|\mathbf{x})} \log \sigma(-\lambda_t \beta T(\sum_{\mathbf{x} \in \mathcal{G}^+} w(\mathbf{x})[L_{\text{dsm}}^\theta(\mathbf{x}, \mathbf{x}_t, \mathbf{c}) - L_{\text{dsm}}^{\theta_{\text{ref}}}(\mathbf{x}, \mathbf{x}_t, \mathbf{c})]$$
$$- \sum_{\mathbf{x} \in \mathcal{G}^-} w(\mathbf{x})[L_{\text{dsm}}^\theta(\mathbf{x}, \mathbf{x}_t, \mathbf{c}) - L_{\text{dsm}}^{\theta_{\text{ref}}}(\mathbf{x}, \mathbf{x}_t, \mathbf{c})])), \tag{17}$$

where $L_{\text{dsm}}^\theta(\mathbf{x}, \mathbf{x}_t, \mathbf{c}) = ||f_\theta(\mathbf{x}_t, t, c) - \mathbf{x}||_2^2$, $L_{\text{dsm}}^{\theta_{\text{ref}}}(\mathbf{x}, \mathbf{x}_t, \mathbf{c}) = ||f_{\theta_{\text{ref}}}(\mathbf{x}_t, t, c) - \mathbf{x}||_2^2$, $\lambda_t$ is a weighting function and the constant $T$ can be factored into $\beta$. We defer the derivation from Eq. (15) to Eq. (17) in the Appendix C. The advantage of the derived DGPO objective is fourfold: 1) *Leverages Relative information*: It directly learns preferences between groups of samples, which leverages the fine-grained relative preference information of individual samples within groups. 2) *Enhances training efficiency*: It does not require training on the entire sampling trajectory, which notably reduces the computational cost per iteration. 3) *Enables effective learning*: It sidesteps the need for an inefficient stochastic policy, thus avoiding inefficient model-agnostic exploration and allowing the model to learn more effectively and directly from the preference data. 4) *Efficient Sampling and Learning*: It allows the usage of deterministic ODE sampling for rollouts. This yields higher-quality training samples compared to inefficient SDE sampling, all while using the same inference budget.

**Timestep Clip Strategy**   The considered online setting requires generating samples from the online model which might be expensive; thus, we take a few steps (e.g., 10) for generating samples to reduce the inference cost following Flow-GRPO (Liu et al., 2025). However, naively performing DGPO's training on these samples generated by few steps would lead to serious performance degradation due to the poor sample quality. To mitigate this, we propose the simple yet effective *Timestep Clip Strategy*: during training, we only sample timesteps from the range $[t_{\min}, T]$ with a chosen minimum timestep $t_{\min} > 0$. This could effectively prevent the model from overfitting specific artifacts (e.g., blurriness) of the generated samples by few steps (see ablation in Fig. 4).

Table 1: **GenEval Result.** We **highlight** the best scores. Obj.: Object; Attr.: Attribution.

| Model | Overall | Single Obj. | Two Obj. | Counting | Colors | Position | Attr. Binding |
|---|---|---|---|---|---|---|---|
| *Autoregressive Models:* | | | | | | | |
| Show-o (Xie et al., 2024) | 0.53 | 0.95 | 0.52 | 0.49 | 0.82 | 0.11 | 0.28 |
| Emu3-Gen (Wang et al., 2024a) | 0.54 | 0.98 | 0.71 | 0.34 | 0.81 | 0.17 | 0.21 |
| JanusFlow (Ma et al., 2025) | 0.63 | 0.97 | 0.59 | 0.45 | 0.83 | 0.53 | 0.42 |
| Janus-Pro-7B (Chen et al., 2025b) | 0.80 | 0.99 | 0.89 | 0.59 | 0.90 | 0.79 | 0.66 |
| GPT-4o (Hurst et al., 2024) | 0.84 | 0.99 | 0.92 | 0.85 | 0.92 | 0.75 | 0.61 |
| *Diffusion Models:* | | | | | | | |
| LDM (Rombach et al., 2022) | 0.37 | 0.92 | 0.29 | 0.23 | 0.70 | 0.02 | 0.05 |
| SD1.5 (Rombach et al., 2022) | 0.43 | 0.97 | 0.38 | 0.35 | 0.76 | 0.04 | 0.06 |
| SD2.1 (Rombach et al., 2022) | 0.50 | 0.98 | 0.51 | 0.44 | 0.85 | 0.07 | 0.17 |
| SD-XL (Podell et al., 2023) | 0.55 | 0.98 | 0.74 | 0.39 | 0.85 | 0.15 | 0.23 |
| DALLE-2 (OpenAI, 2023) | 0.52 | 0.94 | 0.66 | 0.49 | 0.77 | 0.10 | 0.19 |
| DALLE-3 (Betker et al., 2023) | 0.67 | 0.96 | 0.87 | 0.47 | 0.83 | 0.43 | 0.45 |
| FLUX.1 Dev (Labs, 2024) | 0.66 | 0.98 | 0.81 | 0.74 | 0.79 | 0.22 | 0.45 |
| SD3.5-L (Esser et al., 2024) | 0.71 | 0.98 | 0.89 | 0.73 | 0.83 | 0.34 | 0.47 |
| SANA-1.5 4.8B (Xie et al., 2025) | 0.81 | 0.99 | 0.93 | 0.86 | 0.84 | 0.59 | 0.65 |
| SD3.5-M (Esser et al., 2024) | 0.63 | 0.98 | 0.78 | 0.50 | 0.81 | 0.24 | 0.52 |
| w/ Flow-GRPO (Liu et al., 2025) | 0.95 | **1.00** | 0.99 | 0.95 | 0.92 | **0.99** | 0.86 |
| **SD3.5-M w/ DGPO (Ours)** | **0.97** | **1.00** | **0.99** | **0.97** | **0.95** | **0.99** | **0.91** |

Table 2: **Performance on Compositional Image Generation, Visual Text Rendering, and Human Preference** benchmarks. ImgRwd: ImageReward; UniRwd: UnifiedReward.

| Model | Task Metric | | | Image Quality | | Preference Score | | |
|---|---|---|---|---|---|---|---|---|
| | GenEval | OCR Acc. | PickScore | Aesthetic | DeQA | ImgRwd | PickScore | UniRwd |
| SD3.5-M | 0.63 | 0.59 | 21.72 | 5.39 | 4.07 | 0.87 | 22.34 | 3.33 |
| *Compositional Image Generation:* | | | | | | | | |
| Flow-GRPO | 0.95 | — | — | 5.25 | 4.01 | 1.03 | 22.37 | 3.51 |
| **DGPO (Ours)** | 0.97 | — | — | 5.31 | 4.03 | 1.08 | 22.41 | 3.60 |
| *Visual Text Rendering:* | | | | | | | | |
| Flow-GRPO | — | 0.92 | — | 5.32 | 4.06 | 0.95 | 22.44 | 3.42 |
| **DGPO (Ours)** | — | 0.96 | — | 5.37 | 4.09 | 1.02 | 22.52 | 3.48 |
| *Human Preference Alignment:* | | | | | | | | |
| Flow-GRPO | — | — | 23.31 | 5.92 | 4.22 | 1.28 | 23.53 | 3.66 |
| **DGPO (Ours)** | — | — | 23.89 | 6.08 | 4.40 | 1.32 | 23.91 | 3.74 |

## 4 EXPERIMENTS

In this section, we comprehensively evaluate the proposed DGPO. Specifically, we benchmark improvements on three tasks—compositional image generation, visual text rendering, and human preference alignment (Tables 1 and 2). We also present qualitative comparisons and training efficiency (Figs. 2 and 3). We further conduct ablations on key components (Figs. 4 and 5).

### 4.1 EXPERIMENTAL SETUP

**Evaluation Tasks** We evaluate the DGPO on post-training the SD3.5-M (Esser et al., 2024) across three distinct valuable tasks: 1) compositional image generation, using GenEval to test object counting, spatial relations, and attribute binding; 2) visual text rendering (Gong et al., 2025), measuring accuracy of rendering text in generated images, and 3) human preference alignment, using PickScore to assess visual quality and text-image alignment. Details are provided in the Section E.

**Out-of-Domain Evaluation Metrics** To fairly evaluate model performance and guard against reward hacking—where models may overfit to training rewards signal while compromising actual image quality—we employ four independent image quality metrics not used during training as out-of-domain evaluations: Aesthetic Score (Schuhmann et al., 2022), DeQA (You et al., 2025), ImageReward (Xu et al., 2023), and UnifiedReward (Wang et al., 2025). We compute these metrics on DrawBench (Saharia et al., 2022), a comprehensive benchmark featuring diverse prompts.

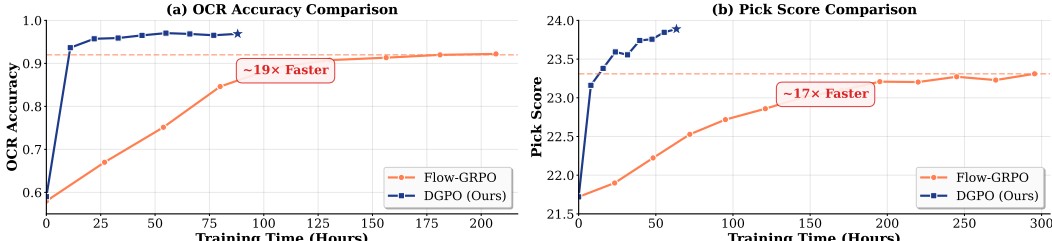

Figure 3: Compare the training speed of Flow-GRPO and our proposed DGPO.

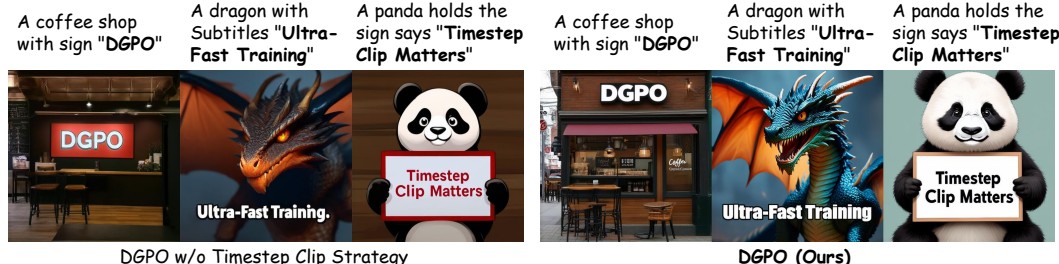

Figure 4: Visual comparisons among variants. It can be seen that without the proposed timestep clip strategy, although it can still accurately follow the instruction, the visual quality notably degrades

## 4.2 MAIN RESULTS

**Quantitative Results**   Table 1 shows that DGPO achieves state-of-the-art performance on GenEval, notably surpassing prior SOTA methods such as GPT-4o and Flow-GRPO. This improvement is achieved while maintaining performance across various out-of-domain metrics (such as AeS, DeQA, and Image Reward), as indicated by Table 2. Beyond compositional image generation, Table 2 provides detailed evaluation results on visual text rendering and human preference tasks, where DGPO similarly demonstrates significant improvements in the target optimization metrics while maintaining performance across various out-of-domain metrics.

**Qualitative Comparison**   We present the qualitative comparisons of methods trained with GenEval's signal in Fig. 2. It is clear that the proposed DGPO can follow the instructions more accurately compared to the base diffusion model and also the Flow-GRPO. Although Flow-GRPO also shows accurate instruction following, its image quality degrades seriously, while our method shows notably better visual quality. We present additional visual samples in Appendix F.

**Training Cost**   The overall training of the proposed DGPO is quick, since we do not require the inefficient stochastic policy for training. Besides, the training of the DGPO is efficient per iteration, since we do not perform training on the whole trajectory. Benefit from these points, the overall training of DGPO for reinforcement post-training is much faster than prior SOTA Flow-GRPO. *As shown in Figs. 1 and 3, the overall training of DGPO is generally around 20× faster than Flow-GRPO.*

## 4.3 ABLATION STUDY

**Effect of Timestep Clip Strategy**   We found that without the proposed timestep clip strategy, the reward metric slightly degrades from 0.96 to 0.95 regarding OCR Accuracy, while the visual quality seriously degrades as shown in Fig. 4.

**ODE Rollout vs. SDE Rollout**   A core advantage of our work compared to prior GRPO-style works (Liu et al., 2025) is the ability to use the efficient ODE solvers for generating samples. This can deliver samples with better quality and rewards. Results in Fig. 5 show that ODE rollout notably outperforms SDE rollout in both convergence speed and ultimate metrics. This suggests that the use

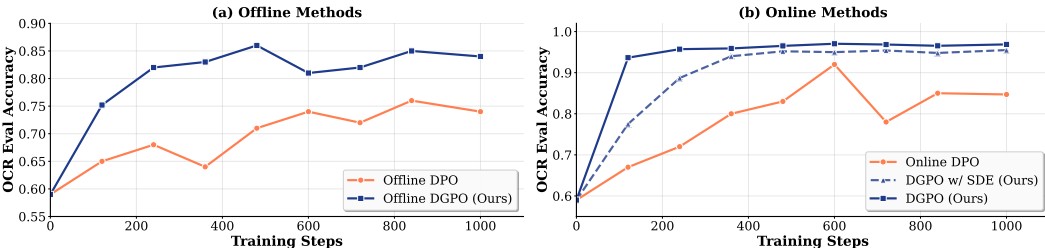

Figure 5: Comparison of visual text rendering across variants.

of SDE in prior works may have been a requirement of the policy gradient framework rather than providing more diverse samples for training.

**Offline DGPO** Our work can be easily adapted to the offline setting by using the reference model $p_{\text{ref}}$ for generating the training dataset. Results in Fig. 5 show that our offline DGPO can reasonably boost the performance over the baseline, but its performance is notably worse than the online setting.

**Compared to Diffusion DPO** Diffusion DPO can also avoid the need for the stochastic policy, however, it cannot leverage the fine-grained reward signals of each sample. Results in Fig. 5 show that our DGPO notably outperforms the DPO in both online and offline settings, indicating the effectiveness of our proposed DGPO in leveraging fine-grained group relative information.

## 5 RELATED WORKS

Recent research has focused extensively on aligning DMs with human preferences through three primary approaches. The first involves fine-tuning diffusion models on carefully curated image-prompt datasets (Dai et al., 2023; Podell et al., 2023). The second approach maximizes explicit reward functions, either by evaluating multi-step diffusion generation outputs (Prabhudesai et al., 2023; Clark et al., 2023; Lee et al., 2023; Ho et al., 2022; Luo et al., 2025a;b) or through policy gradient-based learning (Fan et al., 2024; Black et al., 2023; Ye et al., 2024). The third category employs implicit reward maximization, as demonstrated by Diffusion-DPO (Wallace et al., 2024) and Diffusion-KTO (Yang et al., 2024), which directly leverage raw preference data. A concurrent work (Chen et al., 2025a) explores utilizing the group information in DPO by enumerating all pairwise comparisons within the group. In contrast, our work defines a single group-level reward and reinforces the DMs with maximum-likelihood learning on that group-level reward. Recent works have also adapted GRPO to DMs (Liu et al., 2025; Xue et al., 2025) under policy-gradient framework, demonstrating promising scalability and impressive performance improvements. However, a notable drawback of existing GRPO-style approaches is their reliance on a stochastic policy, which requires inefficient SDE-based rollouts during training. Our work identifies group relative information as the critical component of GRPO and introduces DGPO to directly optimize group preferences, thereby exploiting fine-grained group relative information without requiring stochastic policies. As a result, DGPO achieves significantly faster training and superior performance on both in-domain and out-of-domain reward benchmarks compared to prior GRPO-style methods.

## 6 CONCLUSION

In this work, we introduce Direct Group Preference Optimization (DGPO), a novel online reinforcement learning method specifically designed for post-training diffusion models. Our approach addresses the fundamental mismatch between policy gradient methods like GRPO and the inherent mechanics of diffusion generation. By recognizing that GRPO's effectiveness stems primarily from its utilization of group relative preference information within the group rather than its policy-gradient nature, we developed a method that preserves this key strength while eliminating the need for stochastic policies. DGPO's direct optimization approach offers substantial practical advantages over existing methods. By enabling the use of efficient ODE-based samplers, eliminating reliance on model-agnostic noise for exploration, and avoiding expensive trajectory-based training, DGPO achieves around 20× speedup in overall training time compared to Flow-GRPO. More importantly, our experiments demonstrate that this efficiency gain comes with superior performance, as

DGPO consistently outperforms baseline methods across both in-domain and out-of-domain evaluation metrics.

## ETHICS STATEMENT

This work did not involve human or animal subjects, sensitive data, or any other elements that would necessitate an ethical review. We have identified no potential for misuse or negative societal impact.

## REPRODUCIBILITY STATEMENT

To ensure the reproducibility of our findings, our complete code and experimental setup, which builds upon the open-source Flow-GRPO codebase (Liu et al., 2025), will be made publicly available at `https://github.com/Luo-Yihong/DGPO`.

## ACKNOWLEDGMENTS

Jing Tang's work is partially supported by National Key R&D Program of China under Grant No. 2024YFA1012700, by the National Natural Science Foundation of China (NSFC) under Grant No. 62402410, and by Guangdong Provincial Project (No. 2023QN10X025).

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

# A  USE OF LARGE LANGUAGE MODELS

The use of large language models (LLMs) was strictly limited to language refinement and minor editorial tasks. The authors affirm that LLMs played no part in the substantive phases of the research, which include the ideation, experimental design, data analysis, and the interpretation of results. All scientific content, methodologies, and conclusions presented herein were conceived and developed exclusively by the authors.

# B ADDITIONAL RELATED WORKS

**Diffusion-Based Policies in RL.** Diffusion policies have gained significant traction in recent online RL research. Existing methods have investigated optimizing these policies via different approaches. The first uses reparameterized policy gradients for optimization (Wang et al., 2024b; Ren et al., 2024; Celik et al., 2025); The second explores optimizing via variants of score matching. There are also works exploring optimizing via weighted schemes (Psenka et al., 2023; Yang et al., 2023). These works focus on using diffusion policy for action learning. In contrast, our work explores reinforcing diffusion models in a more diffusion-native way for achieving better performance in the image generation domain.

# C DERIVATION

## C.1 PRELIMINARY: JENSEN'S INEQUALITY.

Let $\phi : \mathbb{R} \to \mathbb{R}$ be a convex function. For a random variable $X$, Jensen's inequality states that the function of the expectation is less than or equal to the expectation of the function:

$$\phi(\mathbb{E}[X]) \leq \mathbb{E}[\phi(X)]. \tag{18}$$

## C.2 DERIVATION OF EQ. 16

For simplicity and without loss of generality, we omit the condition $\mathbf{c}$ in the derivation. The Eq. 15 can be rewritten as:

$$
\begin{aligned}
L(\theta) &= -\mathbb{E}_{(\mathcal{G}^+,\mathcal{G}^-)\sim\mathcal{D}} \log \sigma(\beta[\sum_{\mathbf{x}_0\in\mathcal{G}^+} \mathbb{E}_{q(\mathbf{x}_{1:T}|\mathbf{x}_0)} w(\mathbf{x}_0) \log \frac{p_\theta(\mathbf{x}_{0:T})}{p_{\mathrm{ref}}(\mathbf{x}_{0:T})} \\
&\quad - \sum_{\mathbf{x}_0\in\mathcal{G}^-} w(\mathbf{x}_0)\mathbb{E}_{q(\mathbf{x}_{1:T}|\mathbf{x}_0)} \log \frac{p_\theta(\mathbf{x}_{0:T})}{p_{\mathrm{ref}}(\mathbf{x}_{0:T})}]) \\
&= -\mathbb{E}_{(\mathcal{G}^+,\mathcal{G}^-)\sim\mathcal{D}} \log \sigma(\beta\mathbb{E}_{q(\mathbf{x}_{1:T}|\mathbf{x}_0),\mathbf{x}_0\in\mathcal{G}}[\sum_{\mathbf{x}_0\in\mathcal{G}^+} w(\mathbf{x}_0) \sum_{t=1}^{T} \log \frac{p_\theta(\mathbf{x}_{t-1}|\mathbf{x}_t)}{p_{\mathrm{ref}}(\mathbf{x}_{t-1}|\mathbf{x}_t)} \\
&\quad - \sum_{\mathbf{x}_0\in\mathcal{G}^-} w(\mathbf{x}_0) \sum_{t=1}^{T} \log \frac{p_\theta(\mathbf{x}_{t-1}|\mathbf{x}_t)}{p_{\mathrm{ref}}(\mathbf{x}_{t-1}|\mathbf{x}_t)}]) \\
&= -\mathbb{E}_{(\mathcal{G}^+,\mathcal{G}^-)\sim\mathcal{D}} \log \sigma(\beta\mathbb{E}_{q(\mathbf{x}_{1:T}|\mathbf{x}_0),\mathbf{x}_0\in\mathcal{G}}[\sum_{\mathbf{x}_0\in\mathcal{G}^+} w(\mathbf{x}_0)T\mathbb{E}_t \log \frac{p_\theta(\mathbf{x}_{t-1}|\mathbf{x}_t)}{p_{\mathrm{ref}}(\mathbf{x}_{t-1}|\mathbf{x}_t)} \\
&\quad - \sum_{\mathbf{x}_0\in\mathcal{G}^-} w(\mathbf{x}_0)T\mathbb{E}_t \log \frac{p_\theta(\mathbf{x}_{t-1}|\mathbf{x}_t)}{p_{\mathrm{ref}}(\mathbf{x}_{t-1}|\mathbf{x}_t)}]) \\
&= -\mathbb{E}_{(\mathcal{G}^+,\mathcal{G}^-)\sim\mathcal{D}} \log \sigma(\beta T\mathbb{E}_t\mathbb{E}_{q(\mathbf{x}_t|\mathbf{x}_0),q(\mathbf{x}_{t-1}|\mathbf{x}_t,\mathbf{x}_0),\mathbf{x}_0\in\mathcal{G}}[\sum_{\mathbf{x}_0\in\mathcal{G}^+} w(\mathbf{x}_0) \log \frac{p_\theta(\mathbf{x}_{t-1}|\mathbf{x}_t)}{p_{\mathrm{ref}}(\mathbf{x}_{t-1}|\mathbf{x}_t)} \\
&\quad - \sum_{\mathbf{x}_0\in\mathcal{G}^-} w(\mathbf{x}_0) \log \frac{p_\theta(\mathbf{x}_{t-1}|\mathbf{x}_t)}{p_{\mathrm{ref}}(\mathbf{x}_{t-1}|\mathbf{x}_t)}])
\end{aligned}
\tag{19}
$$

By Jensen's inequality (Section C.1) and the convexity of $-\log\sigma$, we have:

$$
\begin{aligned}
L(\theta) &\leq -\mathbb{E}_{(\mathcal{G}^+,\mathcal{G}^-)\sim\mathcal{D}}\mathbb{E}_{t,q(\mathbf{x}_t|\mathbf{x}_0)} \log \sigma(\sum_{\mathbf{x}_0\in\mathcal{G}^+} w(\mathbf{x}_0)\cdot\beta T\mathbb{E}_{q(\mathbf{x}_{t-1}|\mathbf{x}_t,\mathbf{x}_0)} \log \frac{p_\theta(\mathbf{x}_{t-1}|\mathbf{x}_t)}{p_{\mathrm{ref}}(\mathbf{x}_{t-1}|\mathbf{x}_t)} \\
&\quad - \sum_{\mathbf{x}_0\in\mathcal{G}^-} w(\mathbf{x}_0)\cdot\beta T\mathbb{E}_{q(\mathbf{x}_{t-1}|\mathbf{x}_t,\mathbf{x}_0)} \log \frac{p_\theta(\mathbf{x}_{t-1}|\mathbf{x}_t)}{p_{\mathrm{ref}}(\mathbf{x}_{t-1}|\mathbf{x}_t)}),
\end{aligned}
\tag{20}
$$

## C.3 DERIVATION OF EQ. 17

For simplicity and without loss of generality, we omit the condition $\mathbf{c}$ in the derivation. The DGPO objective in Eq. 16 can be rewritten to:

$$
\begin{aligned}
L(\theta) \leq &-\mathbb{E}_{(\mathcal{G}^+,\mathcal{G}^-)\sim\mathcal{D}}\mathbb{E}_{t,q(\mathbf{x}_t|\mathbf{x}_0)}\log\sigma\big(\sum_{\mathbf{x}_0\in\mathcal{G}^+}w(\mathbf{x}_0)\cdot\beta T\mathbb{E}_{q(\mathbf{x}_{t-1}|\mathbf{x}_t,\mathbf{x}_0)}\log\frac{p_\theta(\mathbf{x}_{t-1}|\mathbf{x}_t)}{p_{\text{ref}}(\mathbf{x}_{t-1}|\mathbf{x}_t)}\\
&-\sum_{\mathbf{x}_0\in\mathcal{G}^-}w(\mathbf{x}_0)\cdot\beta T\mathbb{E}_{q(\mathbf{x}_{t-1}|\mathbf{x}_t,\mathbf{x}_0)}\log\frac{p_\theta(\mathbf{x}_{t-1}|\mathbf{x}_t)}{p_{\text{ref}}(\mathbf{x}_{t-1}|\mathbf{x}_t)}\big),\\
=&-\mathbb{E}_{(\mathcal{G}^+,\mathcal{G}^-)\sim\mathcal{D}}\mathbb{E}_{t,q(\mathbf{x}_t|\mathbf{x}_0)}\log\sigma\big(-\beta T\{\sum_{\mathbf{x}_0\in\mathcal{G}^+}w(\mathbf{x}_0)\cdot[KL(q(\mathbf{x}_{t-1}|\mathbf{x}_t,\mathbf{x}_0)||p_\theta(\mathbf{x}_{t-1}|\mathbf{x}_t))\\
&-KL(q(\mathbf{x}_{t-1}|\mathbf{x}_t,\mathbf{x}_0)||p_{\text{ref}}(\mathbf{x}_{t-1}|\mathbf{x}_t))]-\sum_{\mathbf{x}_0\in\mathcal{G}^-}w(\mathbf{x}_0)\cdot[KL(q(\mathbf{x}_{t-1}|\mathbf{x}_t,\mathbf{x}_0)||p_\theta(\mathbf{x}_{t-1}|\mathbf{x}_t))\\
&-KL(q(\mathbf{x}_{t-1}|\mathbf{x}_t,\mathbf{x}_0)||p_{\text{ref}}(\mathbf{x}_{t-1}|\mathbf{x}_t))]\}\big)
\end{aligned}
\tag{21}
$$

With the Gaussian parameterization (Song et al., 2020) of the posterior $q(\mathbf{x}_{t-1}|\mathbf{x}_t,\mathbf{x}_0) \triangleq \mathcal{N}(\alpha_{t-1}\mathbf{x}_0 + \sqrt{\sigma_{t-1}^2-\eta_{t-1}^2}\frac{\mathbf{x}_t-\alpha_t\mathbf{x}_0}{\sigma_t}, \eta_{t-1}^2 I)$ and reverse sampling $p_\theta(\mathbf{x}_{t-1}|\mathbf{x}_t) \triangleq \mathcal{N}(\alpha_{t-1}f_\theta(\mathbf{x}_t,t) + \sqrt{\sigma_{t-1}^2-\eta_{t-1}^2}\frac{\mathbf{x}_t-\alpha_t f_\theta(\mathbf{x}_t,t)}{\sigma_t}, \eta_{t-1}^2 I)$, the KL divergence term $KL(q(\mathbf{x}_{t-1}|\mathbf{x}_t,\mathbf{x}_0)||p_\theta(\mathbf{x}_{t-1}|\mathbf{x}_t))$ can be computed by:

$$
\begin{aligned}
&KL(q(\mathbf{x}_{t-1}|\mathbf{x}_t,\mathbf{x}_0)||p_\theta(\mathbf{x}_{t-1}|\mathbf{x}_t)\\
&=\frac{1}{2\eta_{t-1}^2}||\alpha_{t-1}\mathbf{x}_0+\sqrt{\sigma_{t-1}^2-\eta_{t-1}^2}\frac{\mathbf{x}_t-\alpha_t\mathbf{x}_0}{\sigma_t}-(\alpha_{t-1}f_\theta(\mathbf{x}_t,t)+\sqrt{\sigma_{t-1}^2-\eta_{t-1}^2}\frac{\mathbf{x}_t-\alpha_t f_\theta(\mathbf{x}_t,t)}{\sigma_t})||_2^2\\
&=\frac{1}{2\eta_{t-1}^2}\left(\alpha_{t-1}-\sqrt{\sigma_{t-1}^2-\eta_{t-1}^2}\frac{\alpha_t}{\sigma_t}\right)^2||\mathbf{x}_0-f_\theta(\mathbf{x}_t,t)||_2^2\\
&=\lambda_t||\mathbf{x}_0-f_\theta(\mathbf{x}_t,t)||_2^2,\quad\text{where }\lambda_t=\frac{1}{2\eta_{t-1}^2}\left(\alpha_{t-1}-\sqrt{\sigma_{t-1}^2-\eta_{t-1}^2}\frac{\alpha_t}{\sigma_t}\right)^2.
\end{aligned}
\tag{22}
$$

Similarly, we have $KL(q(\mathbf{x}_{t-1}|\mathbf{x}_t,\mathbf{x}_0)||p_{\text{ref}}(\mathbf{x}_{t-1}|\mathbf{x}_t) = \lambda_t||\mathbf{x}_0-f_\theta^{\text{ref}}(\mathbf{x}_t,t)||_2^2$. Substituting the computed KL divergence into Eq. 21, we can obtain Eq. 17.

## D  VISUALIZATION OF REWARD HACKING

We set $\beta$ to be smaller (e.g., $\beta = 10$) than its normal value (e.g., $\beta = 100$) and extended the training iterations to make the model over-optimize the rewards. We visualize some failure modes of over-optimizing rewards in Section D.

## E  EXPERIMENT DETAILS

**Compositional Image Generation.** We evaluate text-to-image models on complex compositional prompts using GenEval (Ghosh et al., 2023), which tests six challenging compositional generation tasks including object counting, spatial relations, and attribute binding.

**Visual Text Rendering.** Following the methodology in TextDiffuser (Chen et al., 2023) and Flow-GRPO's experimental setup, we evaluate models' ability to accurately render text within generated images. Each prompt follows the template structure "A sign that says 'text'", where 'text' represents the exact string to be rendered in the image. We measure text fidelity (Gong et al., 2025) as follows:

$$
r = \max(1 - N_e/N_{\text{ref}}, 0)
$$

where $N_e$ denotes the minimum edit distance between rendered and target text, and $N_{\text{ref}}$ represents the character count within the prompt's quotation marks.

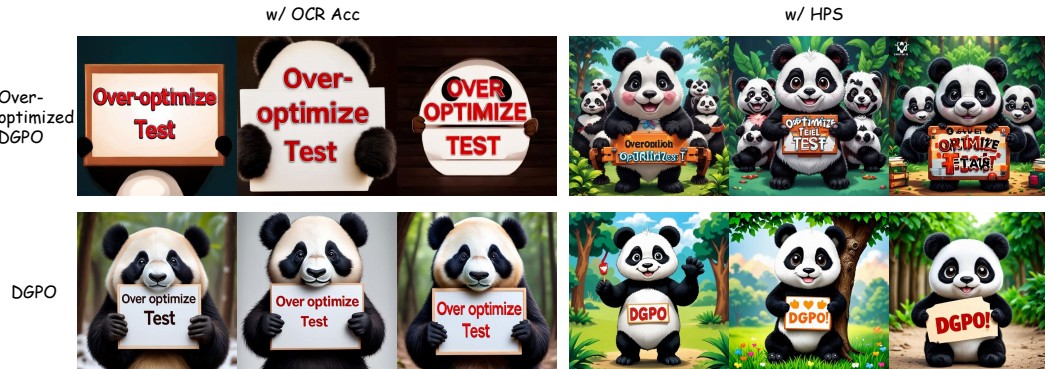

Visualization of reward hacking. Over-optimizing the rule-based reward (i.e., OCR Acc) preserves text accuracy but degrades image quality. In contrast, over-optimizing the model-based reward (i.e., HPS) introduces specific artifacts, such as repeated objects in the background.

**Human Preference Alignment.** To align text-to-image models with human preferences, we employ PickScore (Kirstain et al., 2023) as the reward signal. The PickScore model, trained on large-scale human preference data, evaluates both visual quality and text-image alignment, providing a comprehensive assessment of generation quality from a human-centric perspective.

**Setup Details.** We generate 24 samples for each group for training. We adopt the Flow-DPM-Solver (Xie et al., 2025) with steps of 10 for rollout during training. We adopt LoRA fine-tuning with a rank of 32. The $\beta$ is set to be 100 by default. Defaultly, We update $\theta^-$ by identity mapping, i.e., $\theta^- \leftarrow \theta$ within 200 steps, and update $\theta^-$ by EMA with decay of 0.3 in the remaining training. Experiments are performed over 512 resolution. We use a probability of 0.05 to drop text during training. The experiments are performed over A100. The reported GPU hours are A100 hours.

**Details of the out-of-domain evaluation metrics** We outline the specific out-of-domain metrics used to assess quality: The `aesthetic score` (Schuhmann et al., 2022) employs a linear regression model based on CLIP to evaluate the visual appeal of generated images; For assessing image quality degradation, we utilize the `DeQA score` (You et al., 2025). This metric leverages a multimodal large language model architecture to measure the impact of various imperfections—including distortions, textural degradation, and low-level visual artifacts—on the overall perceived quality of images; `ImageReward` (Xu et al., 2023) serves as a comprehensive human preference model for text-to-image generation tasks. This reward function evaluates multiple dimensions including the coherence between textual descriptions and visual content, the fidelity of generated visuals; Finally, `UnifiedReward` (Wang et al., 2025) represents the latest advancement in this area. This integrated reward framework can evaluate both multimodal understanding and generation tasks, and has demonstrated superior performance compared to existing methods on the human preference assessment leaderboard.

## F ADDITIONAL QUALITATIVE COMPARISON

We present additional visual samples in Figs. 6 and 7.

## G LIMITATIONS AND FUTURE WORKS

Our work focuses on text-to-image synthesis; however, it also has the potential to be adapted to enhance text-to-video synthesis. Exploring the extension would be an interesting future work.

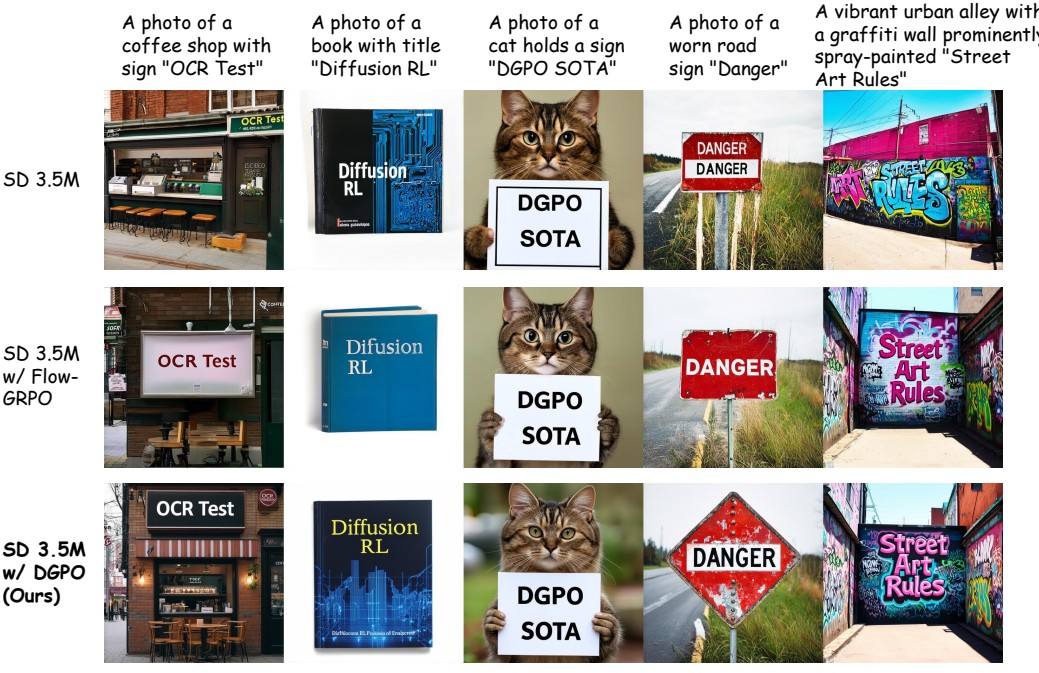

Figure 6: Qualitative comparisons of DGPO against competing methods. The training signal is given by OCR Accuracy. All images are generated by the same initial noise.

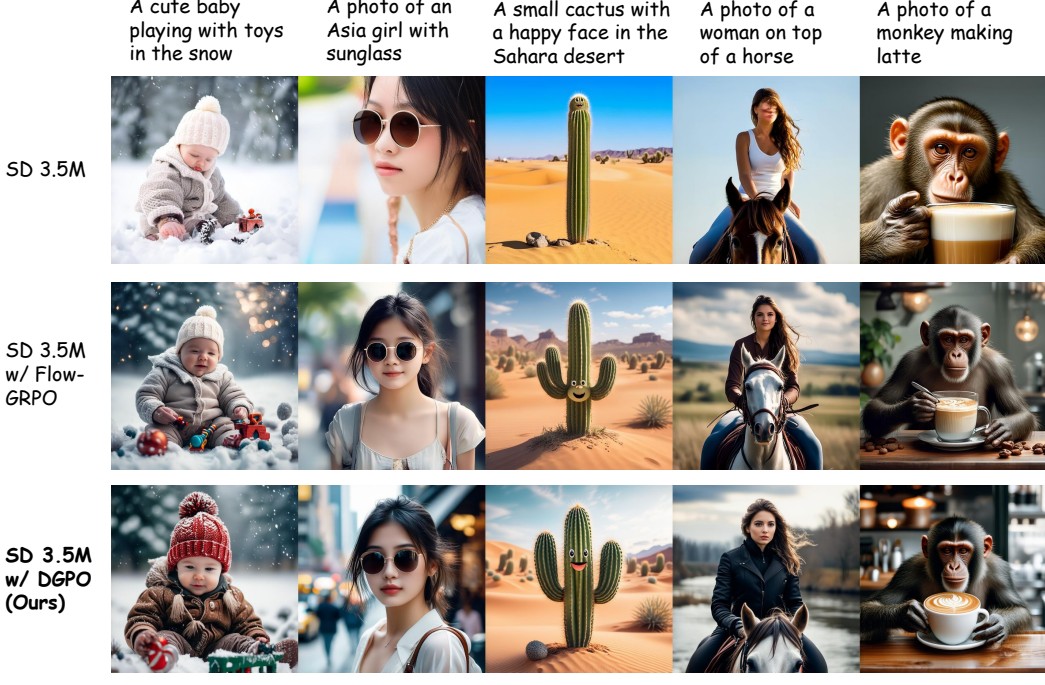

Figure 7: Qualitative comparisons of DGPO against competing methods. The training signal is given by PickScore. All images are generated by the same initial noise.

