# OpenReview forum: "Reinforcing Diffusion Models by Direct Group Preference Optimization"
_ICLR.cc/2026/Conference — ICLR 2026 Poster_

### Official Review · Reviewer_rxkG · 2025-10-25

**Soundness:** 3
**Presentation:** 3
**Contribution:** 3
**Rating:** 6
**Confidence:** 2

**Summary:**

- This paper proposes DGPO, a method that brings GRPO’s within-prompt, group-relative feedback into diffusion alignment, addressing the technical mismatch that has made GRPO difficult for diffusion models.
- The approach distills GRPO’s group information into a tractable objective inspired by diffusion-DPO’s density-ratio formulation, avoiding stochastic policy gradients and full-trajectory training.
- Empirically, DGPO achieves strong preference alignment and image quality compared to existing methods, while remaining computationally efficient.

Note: I used ChatGPT for minor language editing and phrasing assistance; all technical assessments are my own.

**Strengths:**

- The authors propose a novel online RL method for diffusion alignment that efficiently leverages rich group-level information.
- They reformulate the core idea of GRPO—relative, within-prompt group information—and make it applicable to diffusion models using a fast ODE sampler.
- Their DGPO achieves higher benchmark scores than GPT-4o and Flow-GRPO.
- Experiments show DGPO is roughly 15–20× faster than Flow-GRPO at comparable scores.
- They also examine the effectiveness of timestep clipping, ODE rollouts, and the online setting.

Note: I used ChatGPT for minor language editing and phrasing assistance; all technical assessments are my own.

**Weaknesses:**

- My concern is that the authors’ formulation relies on the Bradley–Terry model, which might alter core characteristics of the original GRPO strategy.
- The formulation explicitly partitions each within-prompt group into "good" and "bad" subsets—something the original GRPO did not do—which may change the objective’s behavior compared with the original approach (see my question).

Note: I used ChatGPT for minor language editing and phrasing assistance; all technical assessments are my own.

**Questions:**

- In Equation (17), the DGPO loss contains two separate summations over \(x_0 \in \mathcal{G}^+\) and \(x_0 \in \mathcal{G}^-\).
  Can these be rewritten as a single summation, such as
  $\sum_{x_0 \in \mathcal{G}} A(x_0) (L^{\theta}-L^{\theta_{\mathrm{ref}}})$,
  by using Equation (14)?

- How would you incorporate PPO-style clipping into DGPO? The original GRPO introduced PPO clipping [1].

Note: I used ChatGPT for minor language editing and phrasing assistance; all technical assessments are my own.

---

> ### Author Response · Authors · 2025-11-21
>
> We sincerely thank you for the valuable feedback and for your acknowledgment of our work. We address each comment below.
>
> > My concern is that the authors’ formulation relies on the Bradley–Terry model, which might alter core characteristics of the original GRPO strategy.
>
> We believe the success of GRPO lies in group information, rather than a specific approach for utilization. Our work utilizes the Bradley–Terry model with group information, which is indeed different from policy-gradient based GRPO. Our DGPO  achieves state-of-the-art performance across diverse tasks, which highlights the crucial role of group information.
>
> > In Equation (17), the DGPO loss contains two separate summations over ($x_0 \in \mathcal{G}^+$) and ($x_0 \in \mathcal{G}^-$). Can these be rewritten as a single summation?
>
> Great question. The Equation (17) indeed can be rewritten with single summation to $-E_{(\mathcal{G},c)\sim\mathcal{D}}E_{t,\epsilon}\log \sigma(-\beta T \lambda_t \sum_{x_0\in \mathcal{G}} A(x_0)(L^\theta(x_t,x_0,c) - L^{\theta_{ref}}(x_t,x_0,c)))$.
>
> The current formulation aims to highlight that DGPO's training is based on the Bradley-Terry model to learn the preference between positive and negative groups.
>
> > How would you incorporate PPO-style clipping into DGPO? The original GRPO introduced PPO clipping [1].
>
> Incorporating PPO-style clipping into DGPO is an interesting direction for future work; our current method does not use PPO-style clipping. In DGPO, training is conducted over generated clean data with forward diffusion process, thus, the importance sampling weight should be computed by the likelihood ratio with respect to this clean data, i.e., $\frac{p(x_0)}{p_{old}(x_0)}$.  A possible approach to introduce clipping in this setting would be to replace the density ratio with an ELBO-based ratio, which can be obtained by appropriately reweighting the denoising loss. However, such an approach may introduce additional errors and require extra investigations.
> We will explore the feasibility of this interesting setting in our future work.

---

> > ### Comment · Reviewer_rxkG · 2025-11-26
> >
> > Thank you for response. My concerns about the author's objective function design were resolved. I will keep my positive score.

---

> > > ### Author Response · Authors · 2025-11-28
> > >
> > > Thank you for your reply. We are glad that our response has addressed your concern. Thank you again for your time and effort in reviewing our paper.

---

### Official Review · Reviewer_31Y4 · 2025-10-27

**Soundness:** 2
**Presentation:** 3
**Contribution:** 2
**Rating:** 4
**Confidence:** 4

**Summary:**

The paper presents DGPO, a clear, diffusion-native alternative to GRPO that removes the stochastic-policy requirement while exploiting group-relative preferences.

**Strengths:**

1. The proposed DGPO achieves significant training efficiency improvements and superior performance on benchmark metrics.
2. DGPO integrates the idea of group advantage estimation from GRPO into DPO, thereby avoiding the need for stochastic-policy and eliminating the intractable partition function through an advantage-based weight design.

**Weaknesses:**

1. The mathematical transition from the intermediate objective $\mathcal{L}(\theta)$ (Eq. 15) to the final DGPO training objective $\mathcal{L}_{DGPO}(\theta)$ (Eq. 17) is not explicitly detailed, as the steps involving Jensen's inequality and the convexity of $-\log \sigma$ are only summarized. Try to clarify it.
2. The definition of the term $\log \frac{p _ {\theta}(x _ {t-1}|x _ t,c)}{p _ {ref}(x _ {t-1}|x _ t,c)}$ in $\mathcal{L} _ {DGPO}(\theta)$ is related to the difference in the DSM losses, $L_{dsm}^{\theta}-L_{dsm}^{\theta_{ref}}$ (Eq. 17), but the exact derivation of this equivalence is omitted.
3. The hyperparameter $\beta$ is a critical part of the loss function (Eq. 15, Eq. 17), but its specific value for the main experiments is not provided in the Setup Details, and no ablation study is conducted on its effect, making it difficult to assess the stability and sensitivity of the method to this parameter.
4. The choice of $G=24$ samples per group is mentioned, but the impact of the group size $G$ on the training speed, final performance, and the robustness of the advantage normalization is not investigated.
5. DiffusionNFT [1] also addresses the same problem of Flow-GRPO and has publicly released its source code. It would be beneficial to include a comparison with this method and incorporate the evaluation metrics mentioned in their work.
6. Add related works on diffusion models integrated with online off-policy reinforcement learning, such as QSM [2], DACER [3], and DIME [4], to provide a more comprehensive discussion of recent advancements in this area.

[1] Zheng, Kaiwen, et al. "Diffusionnft: Online diffusion reinforcement with forward process." arXiv preprint arXiv:2509.16117 (2025).

[2] Psenka, Michael, et al. "Learning a diffusion model policy from rewards via q-score matching." arXiv preprint arXiv:2312.11752 (2023).

[3] Wang Y, Wang L, Jiang Y, et al. Diffusion actor-critic with entropy regulator[J]. Advances in Neural Information Processing Systems, 2024, 37: 54183-54204.

[4] Celik, Onur, et al. "Dime: Diffusion-based maximum entropy reinforcement learning." arXiv preprint arXiv:2502.02316 (2025).

**Questions:**

see weakness

---

> ### Author Response · Authors · 2025-11-21
> **Response (1/2)**
>
> We sincerely thank you for the valuable feedback. We address each comment below.
>
> > i) The mathematical transition from the intermediate objective  (Eq. 15) to the final DGPO training objective (Eq. 17) is not explicitly detailed. ii) The derivation of simplification (likelihood to DSM losses) in Eq (17) is omitted.}
>
> Thanks for the valuable suggestion.  The detailed derivation was omitted before because similar derivations have been thoroughly presented in prior work (Diffusion-DPO). However, we recognize that including the complete derivation would improve the self-containedness of our paper. Following your suggestion, we have added the detailed derivations in Appendix C of the revision. Upon careful inspection, we noticed there was a missing expectation sign in Eq. 16. We have fixed this in the revision and made it more clear.
>
> > No ablation study is conducted on its effect, making it difficult to assess the stability and sensitivity of the method to this parameter.
>
> We provide additional experiments on varying $\beta$, and the results are shown below.
>
> $$
> \begin{array}{lccc}
> \hline
> & \text{200 steps} & \text{600 steps} & \text{1200 steps} \newline
> \hline
> \beta = 100 & 0.94 & 0.96 & 0.97 \newline
> \hline
> \beta = 1000 & 0.88 & 0.93 & 0.96 \newline
> \beta = 10 & 0.95 & 0.97 & 0.96 \newline
> \beta = 1 & 0.95 & 0.86 & 0.70 \newline
> \hline
> \end{array}
> $$
>
> The experiments demonstrate that:
>
> 1. Larger values of $\beta$ (e.g., 100 and 1000) lead to more stable optimization but slower convergence.
> 2. Our method is robust across an appropriate range of $\beta$ (from 10 to 1000), consistently achieving strong final performance, with training instability only occurring when $\beta$ is excessively small.
>
> >  impact of the group size on the training speed, final performance, and the robustness of the advantage normalization is not investigated.
>
> We provide additional experiments on varying group size with the overall same batch size, and the results are shown below.
>
> $$
> \begin{array}{lccc}
> \hline
> & \text{200 steps} & \text{600 steps} & \text{1200 steps} \newline
> \hline
> |\mathcal{G}| = 24 & 0.94 & 0.96 & 0.97 \newline
> \hline
> |\mathcal{G}| = 12 & 0.91 & 0.92 & 0.82 \newline
> |\mathcal{G}| = 6 & 0.85 & 0.88 & 0.74 \newline
> \hline
> \end{array}
> $$
>
> The results show that larger group sizes consistently lead to faster convergence, better final performance and improved stability, which again highlights the importance of group information. In contrast, the reduced group size leads to suboptimal performance and unstable training. The phenomenon is also observed in prior works in both the diffusion domain [i] and the LLM domain [ii].
>
> [i] Flow-GRPO: Training Flow Matching Models via Online RL
>
> [ii] ProRL: Prolonged Reinforcement Learning Expands Reasoning Boundaries in Large Language Models

---

> ### Author Response · Authors · 2025-11-21
> **Response (1/2)**
>
> > DiffusionNFT [1] also addresses the same problem of Flow-GRPO and has publicly released its source code. It would be beneficial to include a comparison with this method and incorporate the evaluation metrics mentioned in their work.
>
> Thank you for bringing the interesting concurrent work, DiffusionNFT, to our attention.
> We respectfully note that our work is concurrent with DiffusionNFT; In particular, *DiffusionNFT was submitted to ArXiv on 19 Sep, 2025, which is the same day as the abstract deadline of ICLR 26.* It is hard for us to incorporate a comparison with DiffusionNFT in our initial submission.
>
> We are also glad to provide a discussion below:
> - Our contribution lies in developing a reinforcement learning paradigm that is more diffusion-native, making it more feasible and effective to enhance image generation with RL. Specifically, while GRPO has achieved great success in improving LLM's performance, it demands stochastic policies, which are inefficient for diffusion models. We identify group information as the key to GRPO’s success. Based on this insight, we design DGPO—a method that avoids stochastic policies, is better aligned with diffusion models, and can effectively leverage group information. DGPO achieves SOTA performance while significantly reducing RL post-training costs up to 3\% compared to prior works.
> - In contrast, DiffusionNFT bypasses stochastic policy by extending Negative-aware FineTuning (NFT) to the diffusion domain with group information.
> - Crucially, the fact that two methods developed independently, DGPO and DiffusionNFT, both achieve strong results by removing the policy-gradient framework while retaining group-level information strongly validates the core insight of our paper. The success of DiffusionNFT does not diminish our contribution but rather provides independent corroboration of our core insight.
>
> We also provide a head-to-head comparison with the reported results of DiffusionNFT below.
> $$
> \begin{array} {lccccccccc}
> \hline
> \text{Model} & \text{Iters} & \text{GenEval} & \text{OCR} & \text{PickScore} & \text{ClipScore} & \text{HPSv2.1} & \text{Aesthetic} & \text{ImgRwd} & \text{UniRwd} \newline
> & & (\text{in-domain}) & (\text{in-domain}) & (\text{in-domain}) & (\text{in-domain}) & (\text{in-domain}) & (\text{out-of-domain}) & (\text{out-of-domain}) & (\text{out-of-domain}) \newline
> \hline
> \text{DiffusionNFT} & \text{1.7k} & 0.94 & 0.91 & 23.80 & 0.293 & 0.331 & 6.01 & \mathbf{1.49} & 3.49 \newline
> \text{DGPO (ours)} & \text{1.2k} & \mathbf{0.95} & \mathbf{0.93} & \mathbf{23.85} & \mathbf{0.297} & \mathbf{0.338} & \mathbf{6.12} & 1.46 & \mathbf{3.68} \newline
> \hline
> \end{array}
> $$
> It can be seen that DGPO outperforms DiffusionNFT in most in-domain and out-of-domain metrics.
>
> We believe that the concurrent success of our DGPO and DiffusionNFT will encourage the community to further explore reinforcement learning paradigms that are more natively suited to diffusion models.
>
> > Add related works on diffusion models integrated with online off-policy reinforcement learning, such as QSM [2], DACER [3], and DIME [4], to provide a more comprehensive discussion of recent advancements in this area.
>
> Thanks for the valuable suggestion. We acknowledge that these works represent important recent advances in combining diffusion models with RL. These works primarily focus on learning diffusion policies for continuous actions/control tasks, whereas our work focuses on diffusion RL for image generation.
>
> We also recognize the importance of situating our work within this broader landscape.  We have added a relevant discussion of these methods, along with other related approaches, in Appendix B of the revision. We believe that exploring how such action-oriented diffusion RL methods might be adapted to image generation, or conversely how DGPO could be extended from image generation to action learning, is an exciting direction that we leave for future work.

---

> > ### Comment · Reviewer_31Y4 · 2025-11-23
> >
> > Thank you for your reply. Most of my concerns have been resolved, and I will raise my score to 6.

---

> > > ### Author Response · Authors · 2025-11-24
> > >
> > > Thank you for your reply and for raising the score! We will further include the discussion in our revision following your valuable suggestions. Thank you again for your time and effort in reviewing our paper.

---

### Official Review · Reviewer_Tf9g · 2025-10-30

**Soundness:** 4
**Presentation:** 3
**Contribution:** 4
**Rating:** 8
**Confidence:** 3

**Summary:**

The paper presents an online reinforcement learning method (DGPO), by combining GRPO and DPO. It utilizes the fine-grained preference information from GRPO, but uses ODE-based samplers for efficient training. Once preference groups are constructed, DGPO directly learns from them. Experiments show that DGPO is 20x faster in training, compared with existing methods, and can achieve better performance.

**Strengths:**

- DGPO utilizes group-level preference information.
- DGPO uses ODE-based samplers and therefore trains faster.
- DGPO outperforms Flow-GRPO.

**Weaknesses:**

N/A

**Questions:**

- How to choose group size, and how does it affect the performance?

---

> ### Author Response · Authors · 2025-11-21
>
> We sincerely thank you for the valuable feedback and for your acknowledgment of our work. We address the comment below.
>
> > How to choose group size, and how does it affect the performance?
>
> Thanks for your question. We currently use a group size of 24, following the setting of Flow-GRPO. To better study the impact of group size on performance, we provide additional experiments on varying group size with the overall same batch size.
> $$
> \begin{array}{lccc}
> \hline
> & \text{200 steps} & \text{600 steps} & \text{1200 steps} \newline
> \hline
> |\mathcal{G}| = 24 & 0.94 & 0.96 & 0.97 \newline
> \hline
> |\mathcal{G}| = 12 & 0.91 & 0.92 & 0.82 \newline
> |\mathcal{G}| = 6 & 0.85 & 0.88 & 0.74 \newline
> \hline
> \end{array}
> $$
> The results show that larger group sizes consistently lead to faster convergence, better final performance and improved stability, which again highlights the importance of group information. In contrast, the reduced group size leads to suboptimal performance and unstable training. The phenomenon is also observed in prior works in both the diffusion domain [i] and the LLM domain [ii].
>
> [i] Flow-GRPO: Training Flow Matching Models via Online RL
>
> [ii] ProRL: Prolonged Reinforcement Learning Expands Reasoning Boundaries in Large Language Models

---

### Official Review · Reviewer_3i9L · 2025-11-01

**Soundness:** 3
**Presentation:** 3
**Contribution:** 3
**Rating:** 6
**Confidence:** 4

**Summary:**

This paper introduces Direct Group Preference Optimization (DGPO), a reinforcement learning algorithm to fine-tune diffusion models more efficiently. Current methods are slow because they require inefficient stochastic samplers. DGPO avoids this requirement by learning directly from group-level preferences, which allows the use of efficient deterministic ODE samplers. As a result, DGPO is shown to train up to 30 times faster than existing methods like Flow-GRPO while achieving superior performance on compositional generation, text rendering, and human preference alignment tasks.

**Strengths:**

- Significant Efficiency Gains. The most compelling advantage of DGPO is its dramatic improvement in training efficiency. The claim of being ~20-30 times faster than Flow-GRPO is well-supported by the experiments shown in Figures 1 and 3. This is a substantial practical contribution that could make reinforcement learning-based fine-tuning of diffusion models much more accessible.
- Strong Empirical Performance. DGPO not only trains faster but also achieves state-of-the-art results. It surpasses the performance of the baseline SD3.5-M model and the strong Flow-GRPO competitor on the challenging GenEval benchmark, boosting the score from 63% to 97% (Table 1). The method also shows consistent improvements across visual text rendering and human preference alignment tasks (Table 2).
- Well-Motivated. The paper clearly identifies a key limitation in applying GRPO-style methods to diffusion models (the reliance on inefficient stochastic SDE samplers).
- Comprehensive Experimental Evaluation. The authors provide a thorough evaluation, including qualitative and quantitative comparisons on multiple benchmarks, ablation studies that validate key design choices like the "Timestep Clip Strategy" and the use of ODE rollouts over SDE rollouts, and evaluation on a range of out-of-domain metrics to check for reward hacking.

**Weaknesses:**

- Incremental Novelty. The core idea can be seen as a combination of two existing lines of work: Direct Preference Optimization (DPO) and Group Relative Preference Optimization (GRPO). The main novelty lies in successfully extending DPO to a group-wise setting for diffusion models, but it does not introduce a fundamentally new optimization paradigm. The resulting objective still relies on approximations like Jensen's inequality, making it less theoretically principled and elegant.
- Limited Discussion on Reward Quality. The success of any RL-based fine-tuning heavily relies on the quality of the reward function. The paper uses established reward models like GenEval, OCR accuracy, and PickScore. However, a brief discussion on the potential failure modes or biases of these reward models and how they might affect DGPO's output would strengthen the paper.

**Questions:**

- In "Timestep Clip Strategy," a minimum timestep $t_{\min}$ is used during training to avoid overfitting to artifacts from few-step generation. How sensitive is the final model's performance to the choice of $t_{\min}$? Is there a principled way to select this value, or is it found empirically through hyperparameter tuning?
- How do you cope with Classifier-Free Guidance (CFG) in training and sampling? As I understand, the diffusion loss cannot be applied to the CFG model directly. For example, [1] claims that they abandon CFG entirely during both training and sampling, while still achieving strong performance.

[1] DiffusionNFT: Online Diffusion Reinforcement with Forward Process

---

> ### Author Response · Authors · 2025-11-21
> **Response (1/2)**
>
> We sincerely thank you for the valuable feedback and for your acknowledgment of our work. We address each comment below.
>
> > Incremental Novelty. The core idea can be seen as a combination of two existing lines of work. The main novelty lies in successfully extending DPO to a group-wise setting for diffusion models, but it does not introduce a fundamentally new optimization paradigm.
>
> Our contribution lies in developing a reinforcement learning paradigm that is more diffusion-native, making it more feasible and effective to enhance image generation with RL. Specifically, while GRPO has achieved great success in improving LLM's performance, it demands stochastic policies, which are inefficient for diffusion models. We identify group information as the key to GRPO’s success. Based on this insight, we design DGPO—a method that avoids stochastic policies, is better aligned with diffusion models, and can effectively leverage group information. DGPO achieves SOTA performance while significantly reducing RL post-training costs up to 3\% compared to prior works. ***We believe our work will make Diffusion RL techniques more practical, deliver greater real-world impact, and represent a valuable contribution to the community.***
>
> > The resulting objective still relies on approximations like Jensen's inequality, making it less theoretically principled and elegant.
>
> We believe our method is theoretically principled: it uses the Bradley-Terry model to optimize a group-level reward, performing preference learning between paired groups. To obtain a tractable objective, our method eliminates the intractable normalization constant through advantage-based weighting, and the approximations for further simplifying the objective are consistent with Diffusion-DPO. Specifically, for Jensen's inequality, it obtains a tractable upper bound for training, which is a widely adopted technique in the deep learning field.
>
> > Limited Discussion on Reward Quality. The success of any RL-based fine-tuning heavily relies on the quality of the reward function. The paper uses established reward models like GenEval, OCR accuracy, and PickScore. However, a brief discussion on the potential failure modes or biases of these reward models and how they might affect DGPO's output would strengthen the paper.
>
> We thank the reviewer for this insightful comment. We agree that discussing the limitations of reward models is crucial and will add the following points to our paper.
>
> - Rule-based rewards (e.g., OCR accuracy, GenEval) are generally objective. These rewards strictly evaluate conformance to instructions (like object count or text accuracy) but are agnostic to aesthetic quality. Over-optimizing for these can lead to outputs that satisfy the rule but suffer from degraded aesthetics and composition.
> - Model-based rewards (e.g., HPS, PickScore) approximate subjective human preferences by neural networks. A generator can overfit to generate specific artifact, leading to "reward hacking" -- where outputs achieve high scores but do not genuinely reflect high aesthetic quality or true alignment with user intent.
> - We have included visual samples of the mentioned modes in Appendix D of the revision.
>
> Overall, training based on different rewards may lead to different potential failure modes. This highlights the importance of our evaluation strategy, which assesses performance on a diverse set of out-of-domain rewards. Our DGPO shows strong performance gains across both in-domain and out-of-domain metrics, validates its robustness and the solid improvement.
>
> > In "Timestep Clip Strategy," a minimum timestep is used during training to avoid overfitting to artifacts from few-step generation. How sensitive is the final model's performance to the choice of $t_{min}$? Is there a principled way to select this value, or is it found empirically through hyperparameter tuning?
>
> The selection of $t_{min}$ follows a clear principle: it should correspond to a noise level sufficient to disrupt unwanted low-frequency artifacts (such as blurriness) that emerge in few-step generation, while preserving enough of the denoising steps for effective optimization.
>
> This hyperparameter involves a trade-off. If $t_{min}$ is too small, the model may still overfit to few-step artifacts. Conversely, if $t_{min}$ is too large, optimization becomes difficult as too much of the denoising steps is excluded from training.
>
> Through empirical evaluation, we found that $t_{min}$ exhibits robust performance within the range of [0.3, 0.6]. We typically use \$ t_{min} \in \{ 0.3, 0.4 \} \$, both of which consistently achieve strong performance across diverse tasks. The relatively wide effective range suggests that the method is not overly sensitive to this choice, making it easy to tune in practice.

---

> > ### Author Response · Authors · 2025-11-21
> > **Response (2/2)**
> >
> > > How do you cope with Classifier-Free Guidance (CFG) in training and sampling? As I understand, the diffusion loss cannot be applied to the CFG model directly. For example, [1] claims that they abandon CFG entirely during both training and sampling, while still achieving strong performance.
> >
> > Thanks for the great question. We agree that CFG is not applicable to standard diffusion losses, so **we indeed do not use CFG during training**. Instead, CFG is only applied during sampling. To enable CFG during sampling, we randomly drop text conditioning 5\% of the time during training, following standard diffusion model practice. Thanks for noting the concurrent work of DiffusionNFT and its interesting experimental phenomenon. We also conduct an experiment for our DGPO that drops CFG in both training and sampling.
> >
> > $$
> > \begin{array} {lcccccc}
> > \hline
> > \textbf{Model} & \textbf{OCR Acc.} & \textbf{AeS} & \textbf{DeQA} & \textbf{ImgRwd} & \textbf{PickScore} & \textbf{UniRwd} \newline
> > & \text{(In-domain)} & \text{(Out-of-domain)} & \text{(Out-of-domain)} & \text{(Out-of-domain)} & \text{(Out-of-domain)} & \text{(Out-of-domain)} \newline
> > \hline
> > \text{SD3.5-M} & 0.59 & 5.39 & 4.07 & 0.87 & 22.34 & 3.33 \newline
> > \hline
> > \textbf{DGPO w/o CFG} & 0.95 & 5.30 & 4.02 & 0.97 & 22.38 & 3.43 \newline
> > \textbf{DGPO} & 0.96 & 5.37 & 4.09 & 1.02 & 22.52 & 3.48 \newline
> > \hline
> > \end{array}
> > $$
> >
> > The results show that DGPO without CFG also works well. We hypothesize this is because RL methods can sharpen the sampling distribution of diffusion models, making the distribution more concentrated. As discussed in the autoguidance [a], CFG works by narrowing the distribution and pushing samples toward higher-density regions. We believe that RL operates on a similar principle, effectively playing the role of "CFG distillation."
> >
> > Additionally, we found that although without CFG can perform well on task metrics, it shows notable degradation on out-of-domain metrics. Therefore, we believe that the current implementation of DGPO -- training without CFG and inference with CFG -- represents a better trade-off between performance and computational cost.
> >
> > [a] Guiding a Diffusion Model with a Bad Version of Itself, NeurIPS 2024.

---

> > > ### Comment · Reviewer_3i9L · 2025-11-26
> > >
> > > Thank the authors for the detailed response. I will keep my positive score.

---

> > > > ### Author Response · Authors · 2025-11-28
> > > >
> > > > Thank you for your reply. Thank you again for your time and effort in reviewing our paper.

---

### Meta-Review · Area_Chair_cGLW · 2026-01-12

**Summary:**

The reviewers acknowledged the paper’s strengths: a clear motivation (addressing inefficiency of stochastic SDE samplers in diffusion RL), significant empirical gains (20–30× faster training and strong benchmark performance), and a well-executed evaluation. Some concerns included:

Reviewer 31Y4 requested detailed derivations for the transition and the definition of the advantage term.

Reviewers 3i9L and 31Y4 questioned the choice of the timestep-clip threshold and the scaling parameter, and asked for ablation studies.

Reviewers Tf9g and 31Y4 inquired about the effect of group size on performance and stability.

**Reviewer Concerns:**

The authors’ rebuttal addressed almost all concerns.

**Reviewer Scores:**

Reviewer 3i9L: Initially 6. Estimated final score: 6.

Reviewer Tf9g: Initially 8. Estimated final score: 8.

Reviewer 31Y4: Initially 4. Estimated final score: 6.

Reviewer rxkG: Initially 6. Estimated final score: 6.

---

### Decision · Program_Chairs · 2026-01-26

Accept (Poster)